# Effectiveness of blended learning in pharmacy education: An experimental study using clinical research modules

**Athira Balakrishnan[1], Sreedharan Nair[1], Vijayanarayana Kunhikatta[1], Muhammed Rashid[1], M. K. Unnikrishnan[2,3], P. S. Jagannatha[4], Viji P. Chandran[1], Kanav Khera[1], Girish Thunga[1]***

**1** Department of Pharmacy Practice, Manipal College of Pharmaceutical Sciences, Manipal Academy of Higher Education, Manipal, Karnataka, India, **2** Department of Pharmacy Practice, National College of Pharmacy, Kozhikode, Kerala, India, **3** NGSM institute of Pharmaceutical Sciences, NITTE University, Manglore, Karnataka, India, **4** Statistician, Bangalore, India

\* girish.thunga@manipal.edu

**Data Availability Statement:** All relevant data are within the manuscript and its Supporting Information files.

## Abstract

### Background &objectives

Though there are studies to evaluate the effectiveness of blended learning in pharmacy education, most of them originate from USA and have used previous year students' scores as control. Also there is less research in comparing use of self -regulated learning strategies between blended and other learning strategies. Primary aim was to evaluate the effectiveness of blended learning on knowledge score using clinical research modules. Secondary objective was designed to compare the use of self-regulated learning strategies between blended learning, web-based e-learning and didactic teaching.

### Materials and methods

A prospective cluster randomized trial was conducted with didactic teaching as control and web-based e-learning and blended learning as interventions. The target population was final year Pharm D students. Outcome was assessed using a validated knowledge questionnaire, a motivated strategies for learning questionnaire and a feedback form. All statistical analyses were carried out using Statistical Package for Social Science (SPSS) Version 20.

### Results

A total of 241 students from 12 colleges completed the study. Mean knowledge score of students in blended learning group was higher than those in the didactic teaching and web-based e- learning program (64.26±18.19 Vs 56.65±8.73 Vs 52.11±22.06,p<0.001).Frequency of use of learning strategies namely rehearsal, elaboration, organization and critical thinking was statistically significantly higher in the blended learning group compared to those of didactic and web-based e-learning group (p<0.05) But there were no statistically significant difference of motivational orientations between didactic and blended learning

**Funding:** The authors received no specific funding for this work.

**Competing interests:** The authors have declared that no competing interests exist.

group except strategies of extrinsic goal orientation and self-efficacy. Students preferred blended learning (86.5%) over didactic and web-based e-learning.

## Conclusion

Blended learning approach is an effective way to teach clinical research module. Students of blended learning group employed all motivational and learning strategies more often than students of the didactic and web- based e-learning groups except strategies of intrinsic goal orientation, task value, control of learning belief and help seeking.

## Introduction

Blended learning (BL) or the integration of face-to-face and online instruction is widely adopted across the globe for tertiary education with the increasing use of information technology and electronic learning [1]. There is considerable growth in research analyzing the effectiveness of BL and understanding students' preference and satisfaction towards BL in pharmacy education [2–6]. Some studies have concluded that BL is as effective as traditional teaching [2,3] while a few have reported that BL is more effective than conventional teaching [4–6]. However, most of these studies originate from US, and have employed previous year students' scores as control. Moreover, only few studies come from developing countries. Currently lacking is, prospective controlled trials to analyze the effectiveness of BL in pharmacy education. A systematic review and meta-analysis on this topic is published by same authors [7]. Accordingly, we designed our study to evaluate the effectiveness of BL in final year Pharm D (Doctor of Pharmacy) students employing a 9 hour learning module entitled "Fundamentals of clinical research". "Clinical research" was chosen because India is the preferred destination for multinational companies for clinical trials which raises the demand for skilled qualified personnel. Academia is unable to keep pace with industry because of quick regulatory reforms and technology upgrades thereby making timely syllabus revisions difficult. Training in these areas will help graduates gain more updated knowledge [8,9].

Self-regulated learning (SRL) involves cognitive, metacognitive, motivational, emotional, and behavioural aspects of learning and is currently a subject of extensive research and discussion in educational psychology. Student learning strategies are dynamic and developed to achieve learning objectives in a limited time span [10]. However, there is a little or no study comparing SRL strategies between BL and conventional methods [11]. In this study we have used Pintrich' motivated strategies for learning questionnaire (MSLQ) [12,13] comprised of 15 variables and measures cognitive, metacognitive, motivational, and emotional aspects of learning procedures (Details of MSLQ is provided in S1 File). MSLQ successfully measures motivation and learning strategies in higher education, irrespective of discipline. Accordingly, we designed our secondary objective to compare the use of motivation and learning strategies between BL, web-based e-learning (WEL) and didactic teaching (DT). Blended learning refers to combination of online and face-to face, whereas web-based e-learning refers to all education that takes place online, while didactic teaching refers to all learning experiences that take place face-to-face in the classroom [14–17].

## Materials and methods

### Design and implementation of web-based e-learning platform

**Description of the learning module.** This learning module was intended to teach site-based clinical trial activities to final year Pharm D students. The learning module comprised of five chapters, which included how to start a clinical trial, randomization, informed consent procedure, electronic case report forms, and adverse drug reaction reporting. Each chapter described the theory with the aid of case studies. Details of each chapter are outlined in Table 1. Experts from relevant areas verified contents in each chapter.

### Trial design

A prospective cluster randomized trial, having three arms with DT as control and WEL and BL as interventions 1 & 2 respectively, was conducted between October 2019 to February 2020.

### Ethical consideration

Ethical approval for the study was obtained from Institutional Ethics Committee of Kasturba Hospital Manipal (25/2018), Karnataka, India. This study was registered under clinical trial registry of India (CTRI/2019/11/022021).

### Inclusion criteria for colleges and participants

Colleges affiliated with All India Council for Technical Education (AICTE)/Pharmacy Council of India (PCI) were included. The target population was final year Pharm D students. Students who were interested in participating in the research study were included.

### Recruitment of colleges and participants

Principal investigator approached institutional head through email providing a summary of the study, including randomization, along with an assurance of participant and institutional confidentiality. Details of study was summarized in each email and had informed that colleges

**Table 1. Contents of learning module.**

| Chapter 1: How to start a clinical trial | 1. Introduction<br>2. Confidentiality disclosure agreement, feasibility questionnaire, site evaluation<br>3. Investigators undertaking, financial disclosure agreement and clinical trial agreements<br>4. Site initiation<br>5. IRB-characteristics, composition &reviews<br>6. Reports to be submitted for IRB |
|---|---|
| Chapter 2: Recruitment, Screening &Randomization | 1. Recruitment<br>2. Screening &randomization<br>3. Masking |
| Chapter 3: Informed consent process | 1. Informed consent-general consideration<br>2. How to obtain informed consent |
| Chapter 4: Case report forms | 1.Case report forms<br>2. Paper CRF and eCRF<br>3. How to capture data into eCRF |
| Chapter 5: Reporting of ADR | 1. ADR-common terminologies<br>2. How to report ADR by Principal investigator<br>3. Responsibilities of stakeholders in ADR reporting<br>4. Compensation |

would be allocated to control/interventions based on randomization. Consent was obtained from institutional heads for colleges' participation.

Once consent obtained from institutional heads, investigators visited respective colleges and explained the study to participants. Participant information sheet(PIS) was provided to participants and all interested candidates willfully signed written informed consent form.

### Sample size calculation

Sample size was calculated based on the primary outcome, total knowledge score. Effect size was calculated based on a pilot study, involving 22 students from Manipal College of Pharmaceutical Sciences, Manipal (S2 Appendix for pilot study details). Difference between the largest and smallest means divided by the square root of the mean square error, was used to compute the effect size [18]. A similar effect size was obtained by our meta-analysis of effectiveness of blended learning in pharmacy education [7]. Effect size ($\sigma$) of 0.6, at 5% level of significance, 80% power and design effect (d) of 2, the required sample size was 90 in each method of learning strategies. The required number of colleges in each geographical area was 3, assuming approx. 30 students in each class to each method of learning strategies. Students who participated in pilot study were excluded from the main study.

### Randomization

Colleges that granted permission to conduct the study were grouped based on the geographical areas (GA) namely, Southern Karnataka (GA1), South East Karnataka (GA2), and Kerala (GA3) First, these geographical areas were randomly allocated to three methods of learning strategies and within each geographical area, 4 colleges were selected. There was no random selection of institutions in GA1 as there was only 4 institutions agreed to participate. Eight institutions out of ten agreed to participate were chosen at random in GA2 and GA3 (To reduce the contamination bias, we chose geographical areas as clusters. If students from surrounding colleges enrolled for different arms, there is a chance that they will share their login id & password).

### Study procedure

DT, WEL and BL had the same course contents. All face- to- face sessions in DT & BL were conducted in respective colleges by same experts. Chapters were presented in the format of Microsoft PowerPoint with synchronous voice over slides in website. Script of voice was also attached in platform. Same PowerPoint slides were used to take classes in DT. Website contained 5 chapters, each lasting one hour as well as 4 hours of additional case studies and simulated forms, totaling 9 hours. Each chapter contained theoretical explanation, followed by reinforcing the concept with one or two case studies. Additional case studies and simulated forms were attached to the platform, which was available only to students of WEL. There was a case scenario, based on which students were asked to fill the simulated forms to provide realistic experiences. Simulated form included feasibility questionnaire, ADR reporting form, electronic case report form, and informed consent form. Students of BL were unable to access additional case studies and simulated forms on the website which was taught by face-to-face instruction. Students were required to sign up for first time and log in to the web page to access the learning materials. Students were enrolled for a study period of 8 weeks (details of each learning strategy is depicted in S3 Appendix).

### Web-based e-learning platform for Fundamentals of clinical research

CLINIC E-LEARNING (http://clinilearn.in/my/): A new learning management system, CLIN-ILC E-LEARNING, was developed to implement training program for students. This platform

was developed using Moodle platform Version 3.2. PHP 7.0x was used to code the contents. CLINIC E-LEARNING contained five chapters as described above, and each chapter had pre- and post-test, case study questions, case study for references, important links and files, simulated forms, assignments and feedback forms. Only eligible participants were permitted to access the learning management system through login id and password. (See S4 Appendix for details of Website).

## Experimental and control group

**DT-(control).** Students attended the 9 hour face-to-face lecture delivered by experts with the aid of PowerPoint slides (Same slides of website). Face-to face session was conducted via workshops in each selected college.

**WEL-(Experimental group 1).** WEL group were provided with URL and were asked to sign up with the option to access the website anytime, anywhere. They could clarify doubts by posting their queries to experts in the learner management system. Students were told to complete 9 hour learning module (5 chapters within 6 weeks and remaining two weeks for case study analysis and practicing simulated forms) within 2 months.

**BL-(Experimental group 2).** BL students were provided with 5-hour e-learning and 4 hour face-to-face instruction, totaling 9hours. Students of BL studied the same audiovisual slides (in the same website) during the same period. Followed by, 4 hour discussion by experts (face-to-face instruction provided by the DT team) on additional case studies and important forms used in clinical trials.

## Outcome assessment

We employed the same tests to assess outcomes in all three groups. Outcomes were measured immediately after the completion of the learning module. Outcome assessment consisted of 3 parts (see S5 Appendix for questionnaires). All questionnaires were validated by 5 experts for content validity.

1. Evaluation of knowledge gain: Pre-test and post-test, consisted of same multiple choice questions (50 questions, ten questions from each chapter; 1 mark for each correct answer). Evaluation of conceptual understanding: Case study questions (had 17 items 2 marks for each correct answer).

2. Evaluation of usage of SRL: Motivated strategies for learning questionnaire (MSLQ) by Pintrich et al, comprised of 81 questions. Scoring of MSLQ questionnaire is provided in appendix A.

3. Evaluation of student's experience and satisfaction: Feedback collected in two sessions with 12 questions.

## Statistical analysis

All statistical analyses were carried out using Statistical Package for Social Science (SPSS) Version 20. Continuous variables were reported in terms of mean and standard deviation whereas categorical variables were expressed as percentages. Shapiro Wilks test was performed to ensure the normal distribution of variables. One way Anova or Kruskal Wallis test was conducted to find significant differences in test scores between the groups. Pairwise comparison performed by Dunn's post hoc test, and Games Howell test. Relationships between variables were examined by multiple linear regressions. Spearman correlation was conducted to find

out the correlation between variables. Mann Whitney U test performed to analyze difference in opinions of students of WEL and BL groups.

## Results

A total of 317 students enrolled from 12 colleges, of them 241 students completed the learning module on Fundamentals of clinical research. Only students who finished both pre-test and post-test were included for the analysis.

### Comparison of knowledge score

Pretest scores were compared by one way ANOVA and found that there was no significant difference between groups, $p > 0.05$. As shown in Table 2, mean difference between post-test and pre-test, case studies score, and total score in BL was higher than DT and WEL. There was statistically significant difference in mean difference score (Post to pretest) in knowledge ($p < 0.001$), post-case study score ($p = 0.034$), and total score ($p < 0.001$) between learning strategies. BL showed statistically significant highest total score than didactic teaching ($p < 0.001$) and web-based e-learning program ($p < 0.001$) by Dunn's post hoc test. Results are presented in Table 2.

Multiple linear regression model showed an $R^2$ value of 0.41, which means independent variable (pretest, gender and learning strategies keeping BL as reference category) can explain 41% of the variability of dependent variable (total score). The coefficient on Intervention ($\beta3 = -5.32$) means that, holding all other variables constant, the score on the final test of a student who was taught via the DT is expected to decrease 5.32 points compared to a student who received the BL intervention. Similarly, total score of WEL group students are expected to decrease -12.03 compared to BL group. The findings are depicted in Table 3.

### Comparison of MSLQ scores

It was found that usage of SRL strategies were higher in BL group than DT and WEL except intrinsic goal orientation, task value, control of learning belief and help seeking. There was no statistically significant difference in test anxiety, time management and effort regulation between groups ($p > 0.05$). SRL strategies are composed of 15 variables and were analyzed by Kruskal Wallis test and presented in Table 4.

**Table 2. Comparison of knowledge score between learning strategies.**

| Study Groups | N | Age | Gender (Male: Female) | Pretest (Mean ±SD) | Posttest (Mean ±SD) | Mean Difference (Mean ±SD) | Post case study score (Mean± SD) | Total score |
|---|---|---|---|---|---|---|---|---|
| 1. Control (Didactic teaching) | 87 | 22.43 ±0.58 | 13:74 | 21.82± 5.80 | 33.50±6.63 | 11.67 ± 6.37 | 23.14±5.95 | 56.65 ±8.73 |
| 2. Experimental Group 1(Web -based e-learning) | 62 | 22.61 ±0.49 | 14: 48 | 23.54±11.04 | 30.59±14.12 | 7.04± 8.00 | 21.51±9.80 | 52.11 ±22.06 |
| 3. Experimental Group 2(Blended learning) | 92 | 22.51 ±0.54 | 9:83 | 23.69± 8.82 | 39.39±11.02 | 15.69± 9.88 | 24.87±8.49 | 64.26 ±18.19 |
| F value | | 5.30 | 4.94 | 1.24 | 30.50 | 20.33 | 6.74 | 21.38 |
| p value | | 0.07 | 0.084 | 0.289 | <0.001 | <0.001 | 0.034 | <0.001 |
| Post hoc test<br>  1. DT Vs WEL<br>  2.WEL Vs BL<br>  3. DT Vs BL | | | | | | 1.0.001<br>2.<0.001<br>3.0.004 | 1.0.290<br>2.0.047<br>3. 0.165 | 1. 0.911<br>2. <0.001<br>3<0.001 |

Pretest and mean difference was compared using ANOVA. Post hoc test by Games Howell test as equal variance is not assumed. Kruskall Wallis test for not normally distributed variable (Age, Posttest, Post case study score, total score) and pairwise comparison by Dunn's post hoc test. Gender compared by chi-square test.

**Table 3. Multiple linear regression demonstrating the relationship of total score with other variables.**

| Model variable | Coefficient Estimate | Standard Error | P value |
|---|---|---|---|
| Intercept($\beta_0$) | 36.51 | 3.67 | P<0.01 |
| Pretest($\beta_1$) | 1.18 | 0.01 | P<0.01 |
| Gender($\beta_2$) | -0.43 | 2.41 | 0.858 |
| Didactic teaching(DT) ($\beta_3$) | -5.32 | 2.00 | 0.007 |
| Web-based e-learning(WEL) ($\beta_4$) | -12.03 | 2.21 | P<0.01 |

Multiple linear regression with total score as dependent variable. For learning strategies, BL kept as reference level.

Further pairwise comparison between the groups showed that there was no statistically significant difference in usage of intrinsic goal orientation, task value and control of learning belief between DT and BL (p>0.05). Pairwise comparison of usage of rehearsal, elaboration, organization, and critical thinking indicated there was no statistically significant difference between the DT and WEL. Peer learning showed no significant difference between DT and BL, whereas significant difference exist between the DT and WEL in help seeking.

To determine the relationship between motivational and learning strategy use and final score, Spearman correlation was performed separately for DT, WEL and BL groups. Result showed final score had significant weak positive correlation with extrinsic goal orientation (r = 0.213), control of learning belief (r = 0.280) and rehearsal(r = 226) for students of BL. We did not find any significant correlation between final score and MSLQ strategies in DT and WEL.

## Comparison of students' opinion about contents of web-based e- learning platform

It was found that there was no statistically significant difference(p>0.05) in opinions regarding design of module, clarity of explanation, comprehensive coverage of subject matter, and relevancy of hyperlinks between the WEL and BL groups who used the website for learning. Students of BL were asked their preference for learning strategies and found that 86.5%of students prefer BL, 2.7% and 9.5% preferred DT and WEL, respectively. Students responded that they were satisfied with the clinical research modules and accomplished various learning activities. Mann-Whitney test showed no significant difference between students' opinion (p>0.05).

A total of 92.2% students enrolled for didactic teaching responded that they were satisfied with workshops. More than three quarters of DT students agreed that they were satisfied with clarity of explanation and discussion of module lesson, comprehensive coverage of subject matter, consistency of content with subject objectives and syllabus Results are presented in Table 5.

## Discussion

Published literature suggests that BL generates better performance in pharmacy education, but most of such studies used previous year students' score as control [19,20]. To the best of our knowledge, this report will be the first attempt to evaluate the effectiveness of BL in clinical research modules in pharmacy education. Despite the fact that the term "blended learning" is commonly used, there are some uncertainty about what it means. These approaches are still little understood in higher education, and their descriptions in the literature are inconsistent. In higher education, the term "hybrid" is often used synonymously with "blended" learning, "flipped," "online," or "technology-enhanced" learning [15]. A recent article by Kendra

**Table 4. Comparison of MSLQ scores between learning strategies.**

| MSLQ Domain | Teaching method | Mean ±S.D | P value | Pairwise comparison 1. DT Vs WEL 2. WEL Vs BL 3. DT Vs BL |
|---|---|---|---|---|
| Intrinsic goal orientation | Didactic | 5.42±0.92 | 0.016 | 1.p = 0.026 2.p = 0.038 3.p = 0.905 |
| | Web based | 5.00. ± 1.11 | | |
| | Blended | 5.39± 0.95 | | |
| Extrinsic goal orientation | Didactic | 4.49± 1.38 | 0.010 | 1.p = 0.754 2.p = 0.068 3.p = 0.015 |
| | Web based | 4.72± 1.09 | | |
| | Blended | 5.08 ±1.19 | | |
| Task value | Didactic | 5.41 ± 0.90 | 0.010 | 1.p = 0.017 2.p = 0.025 3.p = 1.00 |
| | Web based | 4.99 ± 1.06 | | |
| | Blended | 5.38 ±1.06 | | |
| Control of learning belief | Didactic | 5.47± 0.88 | P<0.01 | 1.p < .001 2.p = 0.001 3.p = 1 |
| | Web based | 4.84. ± 0.89 | | |
| | Blended | 5.39 ±0.92 | | |
| Self-efficacy | Didactic | 4.77 ± 0.87 | P<0.01 | 1.p = 1 2.p = 0.002 3.p<0.001 |
| | Web based | 4.84 ± 0.93 | | |
| | Blended | 5.31± 0.97 | | |
| Test anxiety | Didactic | 4.06± 1.35 | 0.059 | |
| | Web based | 4.38 ±0.97 | | |
| | Blended | 4.53± 1.34 | | |
| Rehearsal | Didactic | 4.61± 1.02 | P<0.01 | 1.p = 1 2.p = 0.001 3.p = 0.002 |
| | Web based | 4.60 ± 1.04 | | |
| | Blended | 5.09±1.07 | | |
| Elaboration | Didactic | 4.79 ±1.14 | P<0.01 | 1.p = 0.495 2.p<0.001 3.p = 0.015 |
| | Web based | 4.69 ±0.90 | | |
| | Blended | 5.22 ±1.00 | | |
| Organization | Didactic | 4.72 ± 1.18 | P<0.01 | 1.p = 0.237 2.p<0.001 3.p = 0.040 |
| | Web based | 4.50 ± 1.08 | | |
| | Blended | 5.14 ±1.08 | | |
| Critical thinking | Didactic | 4.71 ±0.99 | 0.002 | 1.p = 1 2.p = 0.003 3.p = 0.032 |
| | Web based | 4.66 ± 1.01 | | |
| | Blended | 5.10± 1.08 | | |
| Metacognition | Didactic | 4.67 ±0.76 | P<0.01 | 1.p = 0.029 2.p<0.001 3.p = 0.164 |
| | Web based | 4.43 ± 0.59 | | |
| | Blended | 4.89± 0. 77 | | |
| Time | Didactic | 4.32± 0.68 | 0.077 | |
| | Web based | 4.25 ± 0.47 | | |
| | Blended | 4.43± 0. 50 | | |
| Effort regulation | Didactic | 4.22± 1.05 | 0.585 | |
| | Web based | 4.06± 0.52 | | |
| | Blended | 4.25 ± 0.70 | | |
| Peer learning | Didactic | 4.84 ±1.28 | 0.005 | 1.p = 0.016 2.p = 0.007 3.p = 1 |
| | Web based | 4.40 ±1.18 | | |
| | Blended | 4.93 ± 1.08 | | |
| Help seeking | Didactic | 4.6 ± 1.00 | 0.003 | 1. p = 0.002 2.p = 0.091 3.p = 0.507 |
| | Web based | 4.20± 0.79 | | |
| | Blended | 4.40 ± 0.84 | | |

P value calculated using Kruskal Wallis test and pairwise comparison by Dunn's post hoc test.

**Table 5. Students' attitude towards web -based e-learning program.**

| SL. NO: | Questions | Learning strategy | excellent | good | fair | Poor | P value |
|---|---|---|---|---|---|---|---|
| 1 | Design of the modules | Blended learning | 37.9% | 37.9% | 22.7% | 1.5% | 0.252 |
| | | Web-based | 29.8% | 38.3% | 25.5 | 6.4% | |
| 2 | Explanation of purpose, objectives, and grading procedures | Blended learning | 47% | 39.4% | 10.6% | 3% | 0.874 |
| | | Web-based | 51.1% | 27.7% | 8.5% | 12.8% | |
| 3 | Clarity of explanation and discussion of the module lessons. | Blended learning | 40.9% | 42.4% | 9.1% | 7.6% | 0.745 |
| | | Web-based | 42.6% | 31.9% | 17% | 8.5% | |
| 4 | Consistency of content with subject objectives and syllabus | Blended learning | 33.3% | 47% | 18.2% | 1.5% | 0.908 |
| | | Web-based | 38.3% | 36.2% | 17% | 8.5% | |
| 5 | Comprehensive coverage of subject matter | Blended learning | 43.9% | 42.4% | 12.1% | 1.5% | 0.061 |
| | | Web-based | 31.9% | 40.4% | 17% | 10% | |
| 6 | Relevance of hyperlinks (if any) | Blended learning | 36.4% | 36.4% | 16.7% | 9.1% | 0.210 |
| | | Web-based | 21.3% | 42.6% | 14.9% | 12.8% | |

P value calculated using Mann-Whitney test.

Gagnon distinguish blended and hybrid learning based on face-to- face time [16]. Blended learning does not replace face-to-face time but hybrid learning does. To be more specific, hybrid learning reduces seat time in class. As a result, our model will be better suited to hybrid learning. However, we used the term blended learning because literatures describe hybrid learning is a type of blended learning, and a precise definition remains an ongoing conversation [14,15]. In comparison to blended learning, the term hybrid learning might have been more extensively used in practice than research as there are less highly cited articles on hybrid learning. According to Stefan Hrastinski it is important that researchers and practitioners carefully describe what blended learning means to them. He also proposes, researchers should carefully consider while using a more precise, descriptive term as a supplement or replacement for blended learning [15]. So we believe we have adequately addressed this issue.

Though Clinical trials appear to be concentrated in developed nations, highest percentage increase in registered human clinical research, particularly phase 3 trials are increasing in developing countries [21]. So training in these areas have the potential to enhance students' basic understanding of how the industry works. We have shown that BL is more effective than DT and WEL in teaching clinical research modules, which is in agreement with previous reports in similar settings [22,23]. Ample data have shown that case studies are a gratifying and motivational educational tool that extends declarative and procedural knowledge/expertise [24]. We incorporated case studies in all three learning strategies. Both DT and BL used case study discussions as an active learning strategy in the classroom and enhanced student tutor interaction. Though case studies were incorporated on website and students could post their queries in learning management system itself, there were no provision of face–to-face tutor to student interaction for students who participated in WEL. Students who experienced face-to- face discussion (DT & BL) demonstrated better performance than WEL group, underlining the importance of face-to-face interaction in improving the final score.

In blended learning model class time was effectively used for active learning strategies as students had already viewed recorded videos. This allowed more meaningful face-to face interaction, application of course content and also provided an equal educational value as didactic teaching. BL students spent less time in class compared to DT, which echoes the result of a similar study by the US Department of Education on Distance Education as well as another study by Philips JA [25,26].

BL employed motivational and learning strategies more frequently than DT, except in strategies of intrinsic goal orientation, task value, control of learning belief and help seeking. WEL students scored less in all scales of motivational and learning strategies compared to BL. However, DT students employed all motivational and learning strategies more often than WEL students, except in extrinsic goal orientation, self-efficacy and test anxiety. Interestingly, there is no significant difference between DT and BL in the use of motivational strategies, except in extrinsic goal orientation and self-efficacy; suggesting that face-to—face interaction with tutor considerably influences the use of motivational strategies. This result is in accordance with a study by Kassab et al [27]. DT scored highest for intrinsic goal orientation but least for extrinsic goal orientation, possibly because DT students were personally motivated by the tutor.

BL showed statistically significant highest score in all scales of cognitive strategies (rehearsal, elaboration, organization, and critical thinking) than DT and WEL. Possible reason could be, in BL, students are getting more meaningful interaction with tutors as they are familiar with the content before class. This finding is mostly in accord with a study by Ruchan Uz [28] who reported use of blended instruction is more effective than traditional instruction in terms of developing learning strategies. However our results are in contrast to those of Broadbent [11] who reported that online learning students reported more frequent usage of learning strategies than BL.

Another key finding of this study was that no significant difference was found in metacognitive self-regulation score between DT and BL. So we assume that face-to-face interaction with tutors can demonstrate what to learn, when to learn and how to learn which in turn affect on metacognitive approaches such as planning, monitoring, and regulating.

As expected peer learning and help seeking score was less in WEL group. Help seeking was highest in DT. Understandably, online students felt more isolated when deprived of a sense of community. This is in line with previous studies [11,29].

We have shown that only extrinsic goal orientation, control of learning belief and rehearsal strategies had a weak positive correlation with final score of clinical research examination in BL. On the other hand, SRL did not correlate with final score in DT and WEL.

To the best of our knowledge, there was only one study by Broadbent et al [11] who compared the use of SRL between online and blended environment in higher education. Our result contradicts Broadbent, possibly because, we employed the same learning module for all evaluation unlike much more broad based subject range by Broadbent et al. Moreover, online students had an opportunity for one-to-one interaction with peers and instructors in Broadbent's study. Since student's attitude towards course content also determine SRL, Broadbent's results may not apply to our situation.

In addition, the blended-learning approach used in this course was generally well received by the students. These findings align with other studies that report positive student perceptions of BL in pharmacy education [30,31].

There are some inherent limitations to this study. Firstly, as this study was focused on a single topic, our findings may not be extrapolated to other topics of pharmacy or any other healthcare disciplines. Second, this study accessed short term retention of material. Third, we did not assess how many times students logged in to the website and how long they spent to viewing the materials. As stated by previous reported study [26] we could overcome this by stating that students logging in does not necessarily mean they were actively studying the content. The time obtained may have provided time that the students may not have been physically seated in front of the screen watching the e-lectures because the lecture window would remain active as long as they were signed on to the web-based e-learning portal. Fourth, due to the anonymous nature of feedback form, we were not able to link students' satisfaction level and academic achievement.

## Conclusion and future research lines

Blended learning approach, with online and face-to-face instruction is an effective way to teach clinical research module. Results showed students of blended learning group employed all motivational and learning strategies more often than students of didactic and web- based e-learning groups except strategies of intrinsic goal orientation, task value, control of learning belief and help seeking. Also Students prefer blended learning program than didactic and pure online courses.

Blended learning will be the future of the education sector, since online learning cannot replace physical experience at universities. Given the rapid growth of online studies in last decades, and in the wake of COVID 19, it is paramount to evaluate the effectiveness of computer mediated instruction and to understand how students utilise SRL strategies to achieve academic success. There is an urgent need of fine grained studies with solid experimental design to evaluate the use of SRL between various pedagogies. More insights are needed to correlate the use of SRL and academic achievement. Most BL studies, including current research have assessed short-term retention of knowledge, so future research should address long-term retention of knowledge.

## Supporting information

**S1 Appendix. Motivated strategies for learning questionnaire (MSLQ).**
(DOCX)

**S2 Appendix. Pilot study details.**
(DOCX)

**S3 Appendix. Various strategies used for different teaching learning method.**
(DOCX)

**S4 Appendix. Web-based e-learning program.**
(DOCX)

**S5 Appendix. Questionnaires.**
(DOCX)

**S1 File. Students demographics, knowledge score & MSLQ score.**
(XLSX)

**S2 File. Feedback file.**
(XLSX)

## Acknowledgments

Authors also would like to acknowledge Department of Pharmacy Practice, Manipal College of Pharmaceutical Sciences, and Manipal Academy of Higher Education for all the support and facilities for the best possible completion of this study. We also would like to acknowledge Muscle Infotech. Bangalore for the website development.

## Author Contributions

**Conceptualization:** Athira Balakrishnan, Girish Thunga.

**Data curation:** Athira Balakrishnan, Muhammed Rashid, Viji P. Chandran.

**Formal analysis:** Athira Balakrishnan, Vijayanarayana Kunhikatta, P. S. Jagannatha.

**Investigation:** Sreedharan Nair, Vijayanarayana Kunhikatta, Viji P. Chandran, Girish Thunga.

**Methodology:** Athira Balakrishnan.

**Project administration:** Athira Balakrishnan, Muhammed Rashid, Girish Thunga.

**Resources:** Girish Thunga.

**Software:** Girish Thunga.

**Supervision:** Sreedharan Nair, Girish Thunga.

**Validation:** Athira Balakrishnan, Sreedharan Nair.

**Writing – original draft:** Athira Balakrishnan.

**Writing – review & editing:** Sreedharan Nair, Vijayanarayana Kunhikatta, M. K. Unnikrishnan, P. S. Jagannatha, Kanav Khera.

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
