## [Decision Letter · Decision Letter 0]

4 May 2021

PONE-D-21-09098

Effectiveness of blended learning in pharmacy education: An experimental study using clinical research modules

PLOS ONE

Dear Dr. Thunga,

Thank you for submitting your manuscript to PLOS ONE. After careful consideration, we feel that it has merit but does not fully meet PLOS ONE’s publication criteria as it currently stands. Therefore, we invite you to submit a revised version of the manuscript that addresses the points raised during the review process.

We look forward to receiving your revised manuscript.

Kind regards,

Prof. Ritesh G. Menezes, M.B.B.S., M.D., Diplomate N.B.

Academic Editor

PLOS ONE

Journal Requirements:

Reviewers' comments:

Reviewer's Responses to Questions

**Comments to the Author**

1. Is the manuscript technically sound, and do the data support the conclusions?

Reviewer #1: Yes

Reviewer #2: Yes

Reviewer #3: Partly

Reviewer #4: Yes

Reviewer #5: Partly

2. Has the statistical analysis been performed appropriately and rigorously? 

Reviewer #1: I Don't Know

Reviewer #2: Yes

Reviewer #3: Yes

Reviewer #4: Yes

Reviewer #5: Yes

3. Have the authors made all data underlying the findings in their manuscript fully available?

Reviewer #1: Yes

Reviewer #2: Yes

Reviewer #3: No

Reviewer #4: Yes

Reviewer #5: Yes

4. Is the manuscript presented in an intelligible fashion and written in standard English?

Reviewer #1: No

Reviewer #2: No

Reviewer #3: No

Reviewer #4: No

Reviewer #5: No

5. Review Comments to the Author

Reviewer #1: Thank you for the invitation to review this interesting manuscript. My comments are as follows:

Line 108: The spelling of the e-learning platform seems to be wrong.

Lines 119 and 120: Can the IEC of Kasturba Hospital provide ethical approval for a multi-centre study?

There are missing words and other issues with language in many areas. I would recommend copyediting of the manuscript for language and grammar. Punctuation and spacing between words may also need corrections.

The evaluation questionnaires seem to be long, and students may have to spend several minutes to complete the same. This could have been a problem. How much time did it take on an average to complete the questionnaires?

Line 208: The number of students who completed the module is only 76% of those who enrolled. Will this impact the study?

Line 290: The references should be cited as [19,20].

The authors have mentioned several learning strategies and other constructs in the article. A brief explanation of these will be helpful.

It is good that the authors have included different supporting material with the article.

Reviewer #2: After reviewing this manuscript, it became known that this needs some minor corrections before acceptance. Sentences must be reformed with clear elucidation and a more scientific approach. Also, references must be written in accordance to guidelines provided by the journal.

Reviewer #3: A. Is the manuscript technically sound, and do the data support the conclusions?

The research objectives are relevant. However, the following points needs to be addressed:

1. Regarding the study design, it was mentioned that the three interventions were randomly allocated to three geographical areas, and from where colleges were randomly selected. What is the rationale of having only three clusters? Why did the authors select the geographical area, instead of college, as the cluster level? Please provide the information on the number of colleges considered for the random selection of 12 colleges. It would be helpful for the readers if information available on cluster level factors that might influence the results.

2. Regarding sample size, how was the effect size calculated? on which outcome variable?

3. The study compares three types of learning approaches: BL, DT & WEL. In DT, whole 9 hours learning through face to face; in BL, the 5 hrs didactic part offered through online and 4 hrs case studies through face-to-face discussion; while in WEL, whole 9 hours through web based. However, it is rather unclear, what mentioned here as BL is a blended learning, or flipped classroom, or hybrid learning (for me, it’s more like hybrid learning!; authors used these approaches interchangeably though they are not the same)? All except one references in the introduction were about flipped classroom approach. Interpretation and discussion of results needs to be updated accordingly.

B. Has the statistical analysis been performed appropriately and rigorously?

4. Analysis performed as individually randomised trial!

5. For tables, please provide footnote on method of hypothesis testing

6. In table 2, provide the post-hoc comparison for all significant variables

7. Please clarify, what it means by 1, 2 & 3 in post hoc comparison

8. In table 3 regression modelling, how did the variable selection perform? Why only intervention 2 included?

C. Have the authors made all data underlying the findings in their manuscript fully available?

9. Only summary statistics were provided

D. Is the manuscript presented in an intelligible fashion and written in standard English?

10. Needed substantial revision due to typos (e.g. …website contains 5 chapters per hour…[ page 8]), unclear sentences, and grammatical mistakes.

E. Additional comments:

11. Introduction lacks literature review on the research problem, insufficient background information, and needs to be more structured. All except one reference in the introduction were about flipped classroom approach. The second part of both paragraphs in the introduction better fit in the methods section/discussion section. Objectives can be listed at the end.

12. In the methods section, better move the details of web implementation into under “study procedure” section.

13. Discussion needs to be more structured and include relevant references.

Reviewer #4: Thank you for the opportunity to review this paper. Generally, the topic would be of interest to many readers. Below are suggestions to address

1. Introduction - Why Pintrich’ motivated strategies for learning questionnaire (MSLQ) is selected?

2. Outcome Assessment: How did the items in pre-test and post-test generated? Evaluation of students’ experience and satisfaction – is it newly constructed items or adopted from previous study? Validation?

3. Comparison of knowledge score –

-Line 213 - “As shown in table3, mean difference between post-test and pre-test, case studies score, and total score in BL was higher than DT and WEL.”- is it Table 2 or Table 3?

- Case study scores – pre or post?

- Line 224-225 “Students’ pretest score, and whether they were in BL had a significant influence on final score.”- please clarity

4. The paper also includes inconsistencies e.g. decimal points, capital letter/small letter and would benefit from linguistic editing by a native English speaker prior to resubmission.

Reviewer #5: 1. The objective was to evaluate the effectiveness of blended learning on knowledge score using clinical research modules. Assessing knowledge on a specific subject to demonstrate increased competency in that particular field. However, in this study it was assessed only at the basic level, which are insufficient to argue for the value of learning. I would suggest to evaluate the behavior of the students using objective assessment, including direct observation, an interview or a skills demonstration.

2. The effectiveness of a blended learning environment can be assessed through analysing the relationship between student characteristics/background, design features and learning outcomes. It is always better to determine the significant predictors of effectiveness using learning outcomes as dependent variable.

3. The language is ambiguous and didn't reflect the context clearly.

6. PLOS authors have the option to publish the peer review history of their article (what does this mean?). If published, this will include your full peer review and any attached files.

Reviewer #1: **Yes: **Pathiyil Ravi Shankar

Reviewer #2: No

Reviewer #3: **Yes: **Royes

Reviewer #4: No

Reviewer #5: No

---

## [Author Response · Author response to Decision Letter 0]

18 Jun 2021

Reviewer 1:

1. Line 108: The spelling of the e-learning platform seems to be wrong- Corrected.

2. Lines 119 and 120: Can the IEC of Kasturba Hospital provide ethical approval for a multi-centre study?- Kasturba hospital approved protocol and after submitting permission letter from each colleges, Ethics committee allowed to start the study. Study is conducted in different phases. Pilot study conducted in Manipal college of pharmaceutical sciences (IEC of Kasturba hospital affiliated) and appropriate revisions are made based on pilot study. These changes along with permission letter from each colleges again submitted to the IEC for the next phase approval. 

3. There are missing words and other issues with language in many areas. I would recommend copyediting of the manuscript for language and grammar. Punctuation and spacing between words may also need corrections- Corrected

4. The evaluation questionnaires seem to be long, and students may have to spend several minutes to complete the same. This could have been a problem. How much time did it take on an average to complete the questionnaires?- 

On average, 25 mts for MSLQ questionnaire, 1 hour 10 mts for knowledge test. Usually longer questionnaire will have lower response rate in survey. But its an interventional study & we have informed colleges &students about the nature of study. Only interested students from each colleges participated. So we don’t think number of questions will have an impact in our study.

5. Line 208: The number of students who completed the module is only 76% of those who enrolled. Will this impact the study?

Recommended sample size for three arms was 270, out of that 241 completed. So it may not affect the study. Major reason for dropout was students were absent on particular day of exam. 

6. Line 290: The references should be cited as [19,20].-corrected

The authors have mentioned several learning strategies and other constructs in the article. A brief explanation of these will be helpful. –included, see S3 appendix for learning strategies & S1 for MSLQ variables.

It is good that the authors have included different supporting material with the article.

Reviewer 2:

After reviewing this manuscript, it became known that this needs some minor corrections before acceptance. Sentences must be reformed with clear elucidation and a more scientific approach. Also, references must be written in accordance to guidelines provided by the journal.

Manuscript edited and references corrected 

Reviewer 3:

The research objectives are relevant. However, the following points needs to be addressed:

1. Regarding the study design, it was mentioned that the three interventions were randomly allocated to three geographical areas, and from where colleges were randomly selected. What is the rationale of having only three clusters? Why did the authors select the geographical area, instead of college, as the cluster level? Please provide the information on the number of colleges considered for the random selection of 12 colleges. It would be helpful for the readers if information available on cluster level factors that might influence the results.

Included these points in methodology.

We have selected geographical areas as cluster to minimize contamination bias. If students from nearby colleges enrolled for different arms, there are chances of exchanging login ID and Password. So that may affect the study. So we choose different geographical areas . 

A total of 14 colleges agreed to participate the study from three GAs out of 25. In GA1, only four colleges decided to participate, and there was no random selection. Eight institutions were chosen at random from the ten colleges that provided approval for the study in GA2 and GA3

2. Regarding sample size, how was the effect size calculated? on which outcome variable?

Effect size was calculated based on total knowledge score. Effect size was calculated using difference between highest and lowest mean divided by pooled standard deviation). We obtained a value of 0.66 based on our pilot study. But pilot study results were not significant and conducted in small population. 

So we conducted a meta-analysis of blended learning in pharmacy education, and found that pooled effect size of RCT 0.58.

Reference: Balakrishnan A, Puthean S, Satheesh G, M. K. U, Rashid M, Nair S, et al. Effectiveness of blended learning in pharmacy education: A systematic review and meta-analysis. Plos One.2021;16(6): e0252461.

So we choose 0.6 as final effect size (Not much difference between 0.58 and 0.6 as effect size, also we have taken 1 cluster extra to the sample size)

3. The study compares three types of learning approaches: BL, DT & WEL. In DT, whole 9 hours learning through face to face; in BL, the 5 hrs didactic part offered through online and 4 hrs case studies through face-to-face discussion; while in WEL, whole 9 hours through web based. However, it is rather unclear, what mentioned here as BL is a blended learning, or flipped classroom, or hybrid learning (for me, it’s more like hybrid learning!; authors used these approaches interchangeably though they are not the same)? All except one references in the introduction were about flipped classroom approach. Interpretation and discussion of results needs to be updated accordingly.

We agree with you based on definition” hybrid education is a type of blended learning that substitutes out-of-class online learning activities for face-to-face classroom time, means reducing face-face class room time”. But in higher education, the term “hybrid” is often used synonymously with “blended” learning or alongside terms such as “flipped,” “online,” or “technology-enhanced” learning. A recent systematic review on blended learning in higher education also mentioned “BL is frequently used with terms such as integrated, flexible, mixed mode, multi-mode or hybrid learning” 

A recent article by Stefan Hrastinski, on “What Do We Mean by Blended Learning?” mentioned that “it is likely that papers that use more specific terms, such as blended synchronous learning or flipped classroom, can be found among the huge number of papers that fits under the inclusive blended learning umbrella”. So may be because of this most of our references from Introduction belongs to Flipped learning”. Also all these articles are mentioned their learning strategy as blended learning in introduction or discussion. 

The search for a definition for blended learning has been productive, tough, and at times intimidating. The "hybrid learning" was commonly used before the term "blended learning" became popular. The terms blended learning and hybrid learning are often used interchangeably these days. (Graham 2009; Watson 2008). The term hybrid learning might have been more widely adopted in practice than in research, as there are quite few highly cited papers on hybrid learning, as compared with blended learning research. As far as our knowledge, no published literature clearly states difference between blended, flipped and hybrid learning. differences between BFH models remain ambiguous. Dziuban et al. (2018) suggest that blended is the “new normal” in course formats. Linder (2017) argues that the “flipped” model is simply a subset of the hybrid learning model. Cavanagh, Thompson, and Futch (2017) refer to hybrid as blended. The interchange of these terms persists across the literature.

As a result, there is a lot of ambiguity, and literature claims that hybrid learning falls within blended learning, we used the term blended learning. However we have included these points in discussion.

Included Discussion points:

Despite the fact that the term "blended learning" is commonly used, there are some uncertainty about what it means. These approaches are still little understood in higher education, and their descriptions in the literature are inconsistent. In higher education, the term “hybrid” is often used synonymously with “blended” learning, “flipped,” “online,” or “technology-enhanced” learning. A recent article by Kendra Gagnon distinguish blended and hybrid learning based on face to face time. Blended learning does not replace face to face time but hybrid learning does. To be more specific, hybrid learning reduces seat time in class. As a result, our model will be better suited to hybrid learning. However, we used the term blended learning because literatures describe hybrid learning is a type of blended learning, and a precise definition remains an ongoing conversation. In comparison to blended learning, the term hybrid learning might have been more extensively used in practice than research as there are less highly cited articles on hybrid learning. According to Stefan Hrastinski, it is important that researchers and practitioners carefully describe what blended learning means to them. He also proposes, researchers should carefully consider while using a more precise, descriptive term as a supplement or replacement for blended learning. So we believe we have adequately addressed this issue.

Has the statistical analysis been performed appropriately and rigorously?

4. Analysis performed as individually randomised trial!

As we have less cluster, we did cluster level analysis. (Also cluster was fixed)

References: 

1. Dreyhaupt J, Mayer B, Keis O, Öchsner W, Muche R. Cluster-randomized studies in educational research: principles and methodological aspects. GMS journal for medical education. 2017;34(2).

2. Marion K Campbell, Jill Mollison, Nick Steen, Jeremy M Grimshaw, Martin Eccles, Analysis of cluster randomized trials in primary care: a practical approach, Family Practice, Volume 17, Issue 2, April 2000, Pages 192–196, https://doi.org/10.1093/fampra/17.2.192

3. He Y, Lu J, Huang H, He S, Ma N, Sha Z, Sun Y, Li X. The effects of flipped classrooms on undergraduate pharmaceutical marketing learning: A clustered randomized controlled study. PLoS One.2019; 14(4):e0214624.

5. For tables, please provide footnote on method of hypothesis testing-Provided

6. In table 2, provide the post-hoc comparison for all significant variables-Provided

7. Please clarify, what it means by 1, 2 & 3 in post hoc comparison-Corrected

8. In table 3 regression modelling, how did the variable selection perform? Why only intervention 2 included?

Corrected. Keeping BL as reference, we have modified the model. We have done backward selection ( fit a full model and slowly remove terms one at a time, starting with the term with the highest p-value). Though gender had highest p value, we retained gender since we believe gender has potential role in technology enhanced learning. Next highest p value was Age. We removed it from model. All of the students were in same age group (22-24) and were studying final year pharm D. So we could confidently remove age.

Have the authors made all data underlying the findings in their manuscript fully available?

9. Only summary statistics were provided: Excel data sheet is provided.

D. Is the manuscript presented in an intelligible fashion and written in standard English?

10. Needed substantial revision due to typos (e.g. …website contains 5 chapters per hour…[ page 8]), unclear sentences, and grammatical mistake:

Manuscript edited by professional service. 

 11. Introduction lacks literature review on the research problem, insufficient background information, and needs to be more structured. All except one reference in the introduction were about flipped classroom approach. The second part of both paragraphs in the introduction better fit in the methods section/discussion section. Objectives can be listed at the end.

Literature review of this topic is published by same authors and provided reference in introduction. 

A recent article by Stefan Hrastinski, on “What Do We Mean by Blended Learning?” mentioned that “it is likely that papers that use more specific terms, such as blended synchronous learning or flipped classroom, can be found among the huge number of papers that fits under the inclusive blended learning umbrella”. So may be because of this most of our references from Introduction belongs to Flipped learning”. Also all these cited articles are mentioned about blended learning in introduction or discussion. 

We would like to retain second part of both paragraph in introduction. Because that describes why we selected clinical research module and importance of SRL strategies. Discussion modified according to your suggestion.

12. In the methods section, better move the details of web implementation into under “study procedure” section.

Corrected:

13. Discussion needs to be more structured and include relevant references.

Corrected

Reviewer #4: Thank you for the opportunity to review this paper. Generally, the topic would be of interest to many readers. Below are suggestions to address

1. Introduction - Why Pintrich’ motivated strategies for learning questionnaire (MSLQ) is selected?

Motivated strategies for learning questionnaire is most fitting instrument to measure self- regulated learning strategies as it measures cognitive, metacognitive, motivational, emotional aspects of learning procedures. Since this instrument incurred to examine motivational orientation and learning strategy use for students in higher education irrespective of their discipline, we have used this questionnaire to measure self-regulated learning strategies in present study.

Previous research has been conducted to identify most suitable instrument (Systematic search on pubmed and ERIC using keywords self-regulated learning, reflection, questionnaire, instrument and medical or higher education) and found MSLQ is most appropriate instrument and developed for students in tertiary education, regardless of discipline to examine their motivation for learning and their learning strategies 

Refference: Soemantri D, Mccoll G, Dodds A. Measuring medical students’ reflection on their learning: modification and validation of the motivated strategies for learning questionnaire (MSLQ). BMC medical education. 2018; 18(1): 274.

Also this questionnaire is free and publically available. 

2. Outcome Assessment: How did the items in pre-test and post-test generated? Evaluation of students’ experience and satisfaction – is it newly constructed items or adopted from previous study? Validation?

Questionnaire were developed for the study purpose based on the clinical research module. Questionnaire were validated by 5 experts in the field, for content validity (Questionnaires were validated by 5 experts for its appropriateness, relevance and clarity)

3. Comparison of knowledge score –

-Line 213 - “As shown in table3, mean difference between post-test and pre-test, case studies score, and total score in BL was higher than DT and WEL.”- is it Table 2 or Table 3?- Corrected

- Case study scores – pre or post?- post test, corrected

- Line 224-225 “Students’ pretest score, and whether they were in BL had a significant influence on final score.”- please clarity

Corrected. Keeping BL as reference, we have modified the model. We have done backward selection ( fit a full model and slowly remove terms one at a time, starting with the term with the highest p-value). Though gender had highest p value, we retained gender since we believe gender has potential role in technology enhanced learning. Next highest p value was Age. We removed it from model. All of the students were in same age group (22-24) and were studying final year pharm D. So we could confidently remove age.

4. The paper also includes inconsistencies e.g. decimal points, capital letter/small letter and would benefit from linguistic editing by a native English speaker prior to resubmission-corrected

Reviewer #5: 1. The objective was to evaluate the effectiveness of blended learning on knowledge score using clinical research modules. Assessing knowledge on a specific subject to demonstrate increased competency in that particular field. However, in this study it was assessed only at the basic level, which are insufficient to argue for the value of learning. I would suggest to evaluate the behavior of the students using objective assessment, including direct observation, an interview or a skills demonstration.

A systematic review was conducted on the same topic (Pharmacy education) by same authors and we realized that there are limited number of prospective studies. Also most of the studies conducted from developed nations. Also there are very less number of studies to compare SRL usage between learning strategies. As far as we know, this would be first attempt to evaluate the effectiveness of BL in clinical research modules in pharmacy education

We have conducted focus group discussions to explore the experience of students and included in Pilot study. 

We value your suggestions but study was already completed in 2020, submitted closure report to ethic committee and its impossible to get the same students as all are already passed out. 

2. The effectiveness of a blended learning environment can be assessed through analysing the relationship between student characteristics/background, design features and learning outcomes. It is always better to determine the significant predictors of effectiveness using learning outcomes as dependent variable.

We value your suggestions but study was already completed in 2020, and its impossible to get the same students as all are already passed out.

3. The language is ambiguous and didn't reflect the context clearly.- corrected.

---

## [Decision Letter · Decision Letter 1]

26 Jul 2021

PONE-D-21-09098R1

Effectiveness of blended learning in pharmacy education: An experimental study using clinical research modules

PLOS ONE

Dear Dr. Thunga,

Thank you for submitting your manuscript to PLOS ONE. After careful consideration, we feel that it has merit but does not fully meet PLOS ONE’s publication criteria as it currently stands. Therefore, we invite you to submit a revised version of the manuscript that addresses the points raised during the review process.

Please submit your revised manuscript by 13-August-2021. Please include the following items when submitting your revised manuscript:

A 'Response to Reviewers' letter that responds to each point raised by the academic editor and reviewer(s). You should upload this letter as a separate file labeled 'Response to Reviewers'.A marked-up copy of your manuscript that highlights changes made to the original version. You should upload this as a separate file labeled 'Revised Manuscript with Track Changes'.An unmarked version of your revised paper without tracked changes. You should upload this as a separate file labeled 'Manuscript'.

We look forward to receiving your revised manuscript.

Kind regards,

Prof. Ritesh G. Menezes, M.B.B.S., M.D., Diplomate N.B.

Academic Editor

PLOS ONE

Journal Requirements:

Additional Academic Editor Comments:

- Address the comments made by Reviewer #2. Refer to the following website (5th criterion for publication): https://journals.plos.org/plosone/s/criteria-for-publication

Reviewers' comments:

Reviewer's Responses to Questions

**Comments to the Author**

1. If the authors have adequately addressed your comments raised in a previous round of review and you feel that this manuscript is now acceptable for publication, you may indicate that here to bypass the “Comments to the Author” section, enter your conflict of interest statement in the “Confidential to Editor” section, and submit your "Accept" recommendation.

Reviewer #1: All comments have been addressed

Reviewer #2: All comments have been addressed

Reviewer #3: All comments have been addressed

Reviewer #4: All comments have been addressed

2. Is the manuscript technically sound, and do the data support the conclusions?

Reviewer #1: Yes

Reviewer #2: Yes

Reviewer #3: Yes

Reviewer #4: Yes

3. Has the statistical analysis been performed appropriately and rigorously? 

Reviewer #1: Yes

Reviewer #2: Yes

Reviewer #3: Yes

Reviewer #4: Yes

4. Have the authors made all data underlying the findings in their manuscript fully available?

Reviewer #1: Yes

Reviewer #2: (No Response)

Reviewer #3: Yes

Reviewer #4: Yes

5. Is the manuscript presented in an intelligible fashion and written in standard English?

Reviewer #1: Yes

Reviewer #2: Yes

Reviewer #3: Yes

Reviewer #4: Yes

6. Review Comments to the Author

Reviewer #1: The manuscript can be accepted for publication. The authors have adequately addressed my comments provided during an earlier round of review.

Reviewer #2: Check for grammatical and typographical errors and reform sentence with clear elucidation. The Caption of the tables should be revised and value must be written down to the same decimal place throughout the table.

Reviewer #3: (No Response)

Reviewer #4: Thank you for the opportunity to review this paper again. The authors have addressed comments provided.

7. PLOS authors have the option to publish the peer review history of their article (what does this mean?). If published, this will include your full peer review and any attached files.

Reviewer #1: No

Reviewer #2: No

Reviewer #3: **Yes: **Royes Joseph

Reviewer #4: No

---

## [Author Response · Author response to Decision Letter 1]

13 Aug 2021

Dear sir/madam,

Grammatical errors corrected. Sentences reframed with clear elucidation. Table title modified. Decimal places corrected.

References cross checked. No articles retracted.

(Manuscript edited by professional service). Please revert us if you need any additional information.

Thanks & Regards

Girish Thunga

---

## [Editor Report · Decision Letter 2]

17 Aug 2021

Effectiveness of blended learning in pharmacy education: An experimental study using clinical research modules

PONE-D-21-09098R2

Dear Dr. Thunga,

We’re pleased to inform you that your manuscript has been judged scientifically suitable for publication and will be formally accepted for publication once it meets all outstanding technical requirements.

Kind regards,

Prof. Ritesh G. Menezes, M.B.B.S., M.D., Diplomate N.B.

Academic Editor

PLOS ONE

---

## [Editor Report · Acceptance letter]

23 Aug 2021

PONE-D-21-09098R2 

Effectiveness of blended learning in pharmacy education: An experimental study using clinical research modules 

Dear Dr. Thunga:

I'm pleased to inform you that your manuscript has been deemed suitable for publication in PLOS ONE. Congratulations! Your manuscript is now with our production department. 

Kind regards, 

on behalf of

Prof. Dr. Ritesh G. Menezes 

Academic Editor

PLOS ONE